# Elicited Production of Essential Oil with Immunomodulatory Activity in *Salvia apiana* Microshoot Culture

**DOI:** 10.3390/molecules30040815

**Published:** 2025-02-10

**Authors:** Agata Krol, Adam Kokotkiewicz, Bozena Zabiegala, Klaudia Ciesielska-Figlon, Ewa Bryl, Jacek Maciej Witkowski, Adam Bucinski, Maria Luczkiewicz

**Affiliations:** 1Department of Pharmacognosy, Faculty of Pharmacy, Medical University of Gdansk, Gen. J. Hallera Street 107, 80-416 Gdansk, Poland; adam.kokotkiewicz@gumed.edu.pl (A.K.); mlucz@gumed.edu.pl (M.L.); 2Department of Analytical Chemistry, Faculty of Chemistry, Gdansk University of Technology, 11/12 Gabriela Narutowicza Street, 80-233 Gdansk, Poland; bozzabie@pg.edu.pl; 3Department of Pathophysiology, Faculty of Medicine, Medical University of Gdansk, M. Skłodowskiej-Curie 3a Street, 80-211 Gdansk, Poland; klaudia.ciesielska-figlon@gumed.edu.pl (K.C.-F.); ewa.bryl@gumed.edu.pl (E.B.); jacek.witkowski@gumed.edu.pl (J.M.W.); 4Department of Embryology, Faculty of Medicine, Medical University of Gdansk, M. Skłodowskiej-Curie 3a Street, 80-211 Gdansk, Poland; 5Department of Biopharmacy, Faculty of Pharmacy, Collegium Medicum in Bydgoszcz, Nicolaus Copernicus University in Torun, Jagiellonska Street 15, 85-067 Bydgoszcz, Poland; adam.bucinski@cm.umk.pl

**Keywords:** Lamiaceae, elicitation, in vitro cultures, temporary-immersion bioreactor, lymphocytes, flow cytometry, apoptosis

## Abstract

*Salvia apiana* Jepson is an endemic North American species characterized by a rich phytochemical profile including abietane-type diterpenoids, phenolic acids, flavonoids, and thujone-free essential oil (EO). The current study was aimed at increasing EO production in bioreactor-grown *S. apiana* microshoot culture through biotic elicitation using chitosan, ergosterol, and yeast extract (YE). Additionally, the immunomodulatory effects of the major volatile constituent of white sage—1,8-cineole—as well as EOs obtained from both *S. apiana* microshoots and leaves of field-grown plants, were assessed. EOs were isolated via hydrodistillation and analyzed by GC/MS and GC/FID. Biological assays included flow cytometric evaluation of the proliferation and apoptosis rates of human CD4 and CD8 T lymphocytes, obtained from healthy volunteers and subjected to different concentrations of EOs and 1,8-cineole. Elicitation with 100 mg/L YE improved the production of EO in *S. apiana* microshoots by 9.4% (1.20% *v*/*m*). EOs from both microshoots and leaves of field-grown plants, as well as 1,8-cineole, demonstrated dose-dependent anti-proliferative and pro-apoptotic effects on CD4+ and CD8+ T cells. These findings highlight the potential of *S. apiana* microshoot cultures capable of producing EO with significant immunomodulatory activity.

## 1. Introduction

The *Salvia* L. genus (Lamiaceae) comprises approximately 1000 species, many of which are recognized for their significant pharmacological properties, including their anti-inflammatory and immunomodulatory effects. These biological activities are attributed to the abundance of essential oils in sage plants, which exert their effects by modulating key inflammatory signaling pathways, such as NF-κB, MAPK, and cytokine-mediated pathways, as well as regulating reactive oxygen species (ROS) and reactive nitrogen species (RNS) production [1]. A distinctive species within the *Salvia* genus is white sage (*Salvia apiana* Jeps., subg. Audibertia), a perennial aromatic shrub native to the California and Baja California regions of North America [2]. Its unique characteristics, particularly its high essential oil content (up to 4.3%) and absence of neurotoxic thujone, distinguish it from other sage species and underscore its potential as a natural treatment for dementia, autoimmune diseases, and other inflammation-related disorders [3,4]. Furthermore, the abundance of non-volatile compounds such as phenolic acids, including rosmarinic acid, and abietane-type diterpenoids like carnosic acid and carnosol, also contributes to therapeutic activity of *S. apiana* [5].

*S. apiana* is primarily distributed within the California Floristic Province, a globally recognized biodiversity hotspot known for its high concentration of endemic and endangered plant species [2]. High commercialization of white sage for non-medical purposes as bundled smudge sticks, combined with unbalanced harvests, is a concern held by many Native American groups and conservationists [6]. Given the above, there is a real risk of overexploitation of the species eventually limiting its availability to be useful for therapeutic and scientific purposes. However, alternative sources of biomass can be developed by biotechnological means. In our previous research, microshoot cultures of *S. apiana* were established, scaled up to bioreactor cultivations and demonstrated to accumulate essential oil [4].

The aim of the current work was to improve the productivity of the previously developed in vitro system of white sage in terms of essential oil content, and to assess the immunosuppressive activity of *S. apiana* volatiles in vitro, which (if confirmed) might be indicative of their potential use in inflammation-related disorders. The research was conducted in two parts, covering biotechnological experiments and biological activity studies. In order to enhance volatile oil production, microshoots of white sage were subjected to elicitation, which is considered one of the most effective methods of stimulating the secondary metabolism of plant in vitro cultures [7]. Experiments were conducted in a temporary immersion bioreactor (RITA^®^) using biotic elicitors, chitosan, ergosterol, and yeast extract, applied in different concentrations. The efficacy of elicitor treatments was assessed through phytochemical analysis, which included hydrodistillation and the volumetric estimation of essential oil content, followed by GC/FID and GC/MS analysis of volatile fractions. In the second part of the study, the isolated oil samples were assessed for antiproliferative and pro-apoptotic activity in primary cultures of T lymphocytes (CD4+ and CD8+ cells) isolated from the peripheral blood of healthy volunteers. The effects of essential oil samples from in vitro cultured shoots on T cells were assessed by flow cytometry, and the results were compared with those on the activity of volatile fractions isolated from leaves of wild-grown plants, as well as their major component: 1,8-cineole.

## 2. Results

### 2.1. The Influcence of Elicitation on Microshoots’ Growth, Morphology, and Essential Oil Content

In this study, biotic elicitation was performed using elicitors (chitosan, ergosterol and yeast extract) at concentrations corresponding to those previously applied in other studies on in vitro plant cultures producing essential oils [8,9,10,11]. The elicitors were applied at two specific time points, identified through prior growth curve analysis, corresponding to the late exponential growth phase and the onset of the stationary phase [4]. All tested elicitors affected the growth and production of volatile compounds in *S. apiana* microshoots; however, the culture demonstrated resistance to the applied elicitation treatments, with an increase in the ability to accumulate volatile fractions observed only in response to the addition of yeast extract.

#### 2.1.1. The Influence of Elicitation on Microshoots’ Growth and Morphology

In the applied concentration range, the elicitors did not change the microshoots’ morphology as biomasses remained vivid green and juvenile with no sign of vitrification (Figure 1).

The study demonstrated that ergosterol, at the tested concentration of 10 µM, significantly stimulated the growth of *S. apiana* microshoots (Figure 2). The highest biomass accumulation was observed when ergosterol was added on the 18th day. The calculated Gi was 556.6% and the DW was 18.0 g/L, representing approximately a 1.2-fold increase compared to that of the non-elicited culture (control—Gi 460.1%; DW 17.7 g/L). Prolonged exposure to the elicitor resulted in a decrease in Gi values; however, they remained higher than those of microshoots grown in the control group. Inhibition of growth was demonstrated for microshoots supplemented with ethanol for 3 days (positive control), but no such effect was detected during extended treatment with ethanol.

The growth of the chitosan-supplemented microshoots (200 mg/L) was similar (or slightly higher) to that of the control samples. The calculated Gi values ranged from 492.3% (3 days exposure) to 478.7% (7 days treatment). Additionally, no adverse effects on biomass accumulation were observed in the positive control (treatment with an aqueous HCl solution), confirming the suitability of this solvent for preparing the chitosan stock solution.

Elicitation with yeast extract, in the applied concentration range (50–100 mg/L), did not negatively affect or even slightly reduce the growth of *S. apiana* microshoots. Regardless of the supplementation time, the Gi of the YE-supplemented culture (430.3–474.7%) remained at a similar level as that in the control group (460.1%).

#### 2.1.2. The Influence of Elicitation on Microshoots’ Essential Oil Content

Supplementation with 10 µM ergosterol or the corresponding amount of its solvent (ethanol), applied either 3 or 7 days before harvesting, resulted in up to a 2.3-fold decrease in essential oil content in *S. apiana* microshoots (Figure 3). Prolonged exposure to the elicitor led to a greater reduction in essential oil accumulation. Under such conditions, the total concentration of volatiles decreased to 0.46%, compared to 1.10% recorded for the non-elicited culture. Moreover, treatment with ergosterol reduced the yield of oxygenated monoterpenes, particularly 1,8-cineole, simultaneously increasing the levels of sesquiterpene hydrocarbons in the volatile fraction of *S. apiana* (Table 1, Appendix A). The reduction in 1,8-cineole content was even more pronounced when compared to that in the leaves of field-grown white sage.

Chitosan treatment had a minimal impact on essential oil accumulation in *S. apiana* microshoots. The content of volatiles ranged from 0.85% to 1.06%, which was only slightly lower than that in the control sample (1.10%). Chitosan was more effective in promoting essential oil production when introduced at the onset of the stationary growth phase (3 days treatment) compared to during its addition in the earlier exponential phase (7 days of elicitor exposure). It is noteworthy that among the elicitors used, chitosan supplementation resulted in the lowest decrease in 1,8-cineole content in the biomass compared to non-elicited culture, while still promoting an increase in sesquiterpenes, including caryophyllene.

Yeast extract was the most effective in a concentration of 100 mg/L when applied on the 14th day of the experiment, resulting in an increased essential oil yield compared to that of the control group (1.20% vs. 1.10%, respectively). Similarly to other medium modifications, this treatment led to a reduction in the total monoterpene content of the essential oil by approximately 5.5–10.5% relative to that of the non-elicited group, primarily due to a decrease in the percentage share of 1,8-cineole. Monoterpenes, mainly in the form of oxygenated hydrocarbons (40.6–46.5%), represented the dominant group, whereas sesquiterpenes were detected in much lower concentrations, comprising hydrocarbons (12.9–16.4%) and oxygenated derivatives (3.1–4.0%).

### 2.2. Influence of S. apiana Essential Oil and 1,8-Cineole on T Cell Proliferation and Apoptosis

#### 2.2.1. Isolation and GC Analysis of Volatile Fraction of *S. apiana* Used in Biological Study

The essential oil content in leaves of field-grown plants and microshoots of *S. apiana* was determined to be 1.27% and 4.32% (*v*/*m* dry weight), respectively, using hydrodistillation with a Clevenger apparatus [4]. The dominant component of the essential oil, 1,8-cineole, accounted for approximately 72.7% of the volatile fraction in field-grown leaves and 50.1% in the in vitro biomass. A detailed composition of *S. apiana* essential oils is provided in Table 1. The 1,8-cineole used in the study was obtained commercially.

#### 2.2.2. Influence of *S. apiana* Essential Oils and 1,8-Cineole on Cell Proliferation

The experiment revealed changes in the dynamics of CD4+ and CD8+ T lymphocyte proliferation following stimulation with essential oils isolated from both *S. apiana* microshoots and leaves of field-grown plants, as well as with 1,8-cineole (Figure 4). At the highest tested concentration (1/400 dilution), all substances nearly completely inhibited the proliferation of CD4+ (median values: 4.22%, 4.00%, and 4.19%, respectively, compared to the control—27.07%) and CD8+ cells (median values: 3.95%, 3.83%, and 4.02%; control—27.07%) after 72 h of incubation with the biological material. Similar inhibition of T lymphocyte proliferation was observed after 120 h of incubation with the tested volatile terpenes. When comparing the results to those of the negative control, such a strong effect on the lymphocytes was not observed at dilutions of 1/4000 and 1/40,000 of the essential oil fractions. The only exception was the essential oil isolated from *S. apiana* microshoots, tested at a 1/4000 dilution, where the percentage of proliferating CD4+ and CD8+ cells was significantly lower than in the control. In general, as the concentration of the volatile fraction in the culture decreased, the percentage of proliferating cells increased, indicating a dose-dependent effect.

#### 2.2.3. Influence of *S. apiana* Essential Oils and 1,8-Cineole on Cell Death

In the next step, the influence of white sage volatile fractions and 1,8-cineole on the apoptosis of lymphocytes was established. Cytometric analyses demonstrated that the incubation of *S. apiana* essential oil from *S. apiana* leaves of field-grown plants and microshoots, as well as 1,8-cineole, with *T. lymphocytes* induced apoptosis, resulting in a reduction in live and early apoptotic cells, while increasing the proportion of late apoptotic and necrotic cells compared to that of the control (Figure 5). This effect was most pronounced at the highest concentration of volatile oil applied in the experiment (1/400 dilution). After 72 h, the median percentage of alive T lymphocytes was 1.29%, 0.64%, and 54.31%, whereas the control group showed a significantly higher median of 62.61%. The percentage of early apoptotic lymphocytes also decreased (median values: of 3.57%, 2.03%, and 17.14%, compared to the control—25.08%), while the proportion of late apoptotic cells increased (median values of 91.37%, 96.20%, and 25.52% vs. the control—14.62%), as did the rate of necrotic cells (median values: 2.66%, 1.40%, and 0.54% vs. control—0.37%). These effects were less pronounced at dilutions of 1/4000 and 1/40,000 of the essential oils and 1,8-cineole, where the necrotic effects on the T cells were comparable to those observed in the control group. The results indicate a cytotoxic effect of high concentrations of volatile fractions from *S. apiana*, derived from both in vitro cultures and field-grown plants, even after 120 h of exposure.

## 3. Discussion

In this study, the effects of elicitation on the production of essential oil in *S. apiana* in vitro cultures were investigated for the first time. So far, there is a limited number of reports regarding the effects of stress factors on the accumulation of volatile terpenoids in in vitro shoot cultures [8,12,13,14]. Nevertheless, data from the literature indicate that elicitor treatment can affect both the quantity and composition of volatile fractions. The selection of elicitors and their concentrations was based mainly on previous studies on the secondary metabolism of sage cell cultures [15,16]. In the current work, three commonly used biotic elicitors were tested. The experimental conditions were chosen with the understanding that elicitation can negatively affect growth, highlighting the importance of optimizing both the elicitor concentration and the timing of its application. All experiments were conducted using the RITA^®^ temporary immersion reactor, which has previously been demonstrated to be a suitable system for the cultivation of *S. apiana* microshoots [4] and a proper platform for elicitation experiments [8].

Chitosan, yeast extract, and ergosterol, as well as other molecules, such as lipopolysaccharides (LPS), β-glucans, flagellin, and peptidoglycans, are recognized as microbe- or pathogen-associated molecular patterns (M/PAMPs), which are highly conserved molecular structures derived from microorganisms. Recognized by pattern recognition receptors (PRRs) in plants, these M/PAMPs play a critical role in the activation of plant innate immune responses. This process initiates a cascade of defense-related mechanisms, including the biosynthesis of antimicrobial compounds like phytoalexins, as plants respond to perceived pathogen attack. Consequently, these elicitors are widely utilized to mimic pathogen-induced plant defense pathways and stimulate the production of specialized metabolites [17,18,19]. The selection of chitosan and yeast extract from other M/PAMPs was based on their established ability to stimulate terpenoid biosynthesis in sage, especially tanshinones in hairy root cultures of *Salvia* spp. [16]. While ergosterol has been investigated as an elicitor in a limited number of studies, it has shown potential to enhance essential oil production in *Ledum palustre* microshoot culture [8] and sesquiterpenoids in tobacco cells [17]. Additionally, ethanol, used as a solvent for the ergosterol stock solution, may itself act as a stimulant of secondary metabolism in plants [8,20,21,22].

As presented in the results section, only yeast extract, at a concentration of 100 mg/L, and exposure time of 7 days, improved essential oil production in *S. apiana* microshoots. Under such conditions, the volatile content was 9.4% higher than the non-elicited biomass; however, the recorded increase was not statistically significant. As far as other experimental variants are concerned, none of them were effective in stimulating essential oil production, with volatile contents comparable to those of the control group. In contrast, results from other studies on in vitro cultures of *Salvia* spp. have demonstrated that elicitation strategies were effective in stimulating secondary metabolism when applied to phenolic acids, as abiotic elicitors upregulated key genes involved in their biosynthetic pathways, leading to an increased accumulation of these compounds in the investigated biomasses [23]. However, data from the literature also indicate that the specific response to elicitation can vary depending on *Salvia* species and culture type (e.g., microshoots, hairy roots, suspension cells) [15,16]. Therefore, the limited response to elicitation observed in *S. apiana* microshoots may be either due to species-specific traits or the particular culture type used. Given the lack of data regarding in vitro cultures of white sage, further research in this field is needed, both to understand the underlying mechanisms of the stress response (or lack thereof), and to explore alternative elicitation strategies that might be more effective for enhancing essential oil production in *S. apiana*. In particular, considering the limited efficacy of yeast extract in this regard, it is reasonable to investigate its effects in a broader concentration range. Another approach could involve the use of certain heavy-metal salts as elicitors, since some reports indicate that they may be effective in stimulating/diverting the secondary metabolism of sage plants. For instance, silver ions (combined with yeast extract) proved to be exceptionally effective in stimulating tanshinone biosynthesis in hairy root cultures of *Salvia miltiorrhiza* [24]. Studies on field-grown aromatic plants (including representatives of the mint family) also indicate that heavy metal stress can affect both the composition and quantity of essential oil [25]. It is thus worth investigating whether the observed effects can be reproduced in a controlled manner under in vitro conditions.

Despite the ineffectiveness of elicitor treatments in terms of increasing essential oil content, the tested substances induced a significant reduction in the relative content of monoterpene hydrocarbons, particularly 1,8-cineole, in the volatile fraction of *S. apiana* microshoots. This was associated with a notable increase in the abundance of sesquiterpene hydrocarbons, primarily caryophyllene and aristolene. In contrast to our findings, studies on *Mentha* × *piperita* shoots have demonstrated an increase in oxygenated monoterpenes following elicitation, highlighting species-specific differences in the metabolic response to elicitors. Specifically, salicylic acid (50 µM) and copper sulfate (25 µM) treatments resulted in a marked increase in oxygenated monoterpenes, while chitosan (200 mg/L) elicitation resulted in less marked alterations between monoterpene hydrocarbons and oxygenated monoterpenes [9]. Additionally, under both natural and greenhouse conditions, elicitation techniques have been widely demonstrated to induce changes in the composition of volatile fractions in aromatic plants [26,27,28]. However, in *S. apiana* microshoot cultures, including those treated with elicitors, the composition of essential oil remained consistent with that typically observed in this species [3]. No novel compounds or significant alterations beyond previously documented volatile profiles were detected, suggesting that the biosynthetic pathways in this culture system were not substantially redirected by the applied treatments. A notable exception is ferruginol, the presence of which in essential oil from both *S. apiana* microshoots and an intact plant was also reported in our previous work [4]. Ferruginol belongs to abietane-type diterpenoids and is commonly found in numerous sage species, including *S. apiana* [29]. However, apart from our studies, ferruginol has not been previously reported in white sage essential oil [3]. A similar discrepancy was also reported for *Salvia officinalis*, with studies either confirming [30] or failing to confirm its presence [3,31] in the volatile fraction. This can obviously result from the variability of the plant material studied; however, the differences in protocols of essential oil isolation and analysis are also likely a contributing factor. Being a diterpene, ferruginol is substantially less volatile than monoterpene and sesquiterpene constituents of sage, and thus is more difficult to isolate via hydrodistillation. Depending on the distillation conditions, ferruginol may be extracted to a different extent, which may explain the previously mentioned discrepancies in the reports. Consequently, we cannot assume that it was exhaustively isolated in the current work. Nevertheless, the discussed compound may contribute to the bioactivity of the volatile fraction since it exhibits cardioactive, antihypertensive and cytotoxic effects [29]. Most importantly, however, the present work demonstrated that both field-grown *S. apiana*, as well as its microshoots obtained by in vitro culture, produce complex essential oils in quantities allow for their isolation and testing. Based on our previous observations of the proliferation- and apoptosis-modifying effects of essential oils isolated from *Rhododendron tomentosum* [32] and *Nigella sativa* [33], we tested if the essential oils from *S. apiana* have the similar effects and found this working hypothesis to be true. Detailed chemical analysis of the *S. apiana* essential oil compositions indicated that they contain multiple components, some of which (e.g., α-pinene) were also demonstrated to cause an earlier decrease in the proliferation of human leukemic and lymphomatic T cells [34,35]. Apart from potential antileukemic/antilymphomatic activities, the properties of *S. apiana* essential oil may—in our opinion—potentially lead to their clinical testing and then therapeutic use in chronic diseases requiring some level of immunosuppression (e.g., autoimmune diseases including the rheumatoid arthritis).

Additionally, it is noteworthy that the tested samples of *S. apiana* essential oils exhibited significant inhibitory effects on acetylcholinesterase, tyrosinase, and hyaluronidase [4], indicating a broad spectrum of biological activities inherent to the monoterpenes and sesquiterpenes present in *S. apiana*. These properties, including immunosuppressive effects, underscore the potential of *S. apiana* as a herbal drug in modern phytotherapy, particularly for autoimmune and inflammatory disorders [33,36,37,38]. Furthermore, the findings suggest that high-quality essential oil can be produced through biotechnological methods, reducing dependence on the limited natural resources of this species.

It should be emphasized that 1,8-cineole is used as an active substance with anti-inflammatory, mucolytic, and bronchodilatory properties under the trade name Soledum Forte^®^ [39,40,41]. Despite the cytotoxic activity observed against T lymphocytes in our study, 1,8-cineole is generally well tolerated and exhibits a favorable safety profile [41]. Numerous in vitro, in vivo, and clinical studies have demonstrated the beneficial effects of 1,8-cineole, including inhibiting pro-inflammatory cytokines, such as TNF-α, IL-1β, and IL-6, while promoting the expression of anti-inflammatory mediators like IL-10 [39,41,42,43]. A study by Juergens and co-workers [43] demonstrated the inhibitory activity of 1,8-cineole on TNF-a and IL-1b in cultured human lymphocytes and monocytes. Besides its effects on the immune system, 1,8-cineole has also been shown to target cancer cells; studies have shown that both 1,8-cineole and the essential oil of *Achillea membranacea* Labill., which contains 30.4% 1,8-cineole, demonstrated cytotoxic activity against A2780 ovarian cancer cells, inducing apoptosis and inhibiting cell proliferation [44]. This cytotoxic activity has also been observed in colon cancer cells, where 1,8-cineole induced apoptosis and inhibited cell proliferation [45]. However, to the best of our knowledge, no studies to date have investigated its antiproliferative and pro-apoptotic effects on human T cells.

## 4. Materials and Methods

### 4.1. Plant Materials

The use of plant material in the study complies with relevant institutional, national, and international guidelines and legislation. Plant species identification was performed by Marcin Gorniak and Aleksandra M. Naczk in previous research [4]. In the study, the previously established in vitro microshoot culture of *Salvia apiana* Jepson served as a source of plant material for biotechnological experiments. The culture was maintained on Schenk–Hildebrandt (SH) medium [46], supplemented with 0.22 mg/L thidiazuron (TDZ), 2.0 mg/L 6-(γ,γ-dimethylallylamino)purine (2iP), 3.0% (*w*/*v*) sucrose and 0.6% (*w*/*v*) agar.

The essential oils of microshoots and field-grown leaves of *S. apiana*, used in the biological experiments, were isolated and analyzed in the previous research [4]. Briefly, microshoots grown on solid SH medium, supplemented with 0.22 mg/L TDZ, 2.0 mg/L 2iP, 3.0% (*w*/*v*) sucrose and 0.6% (*w*/*v*) agar, were collected and dried at 30 °C for 24 h, while commercially available dried leaves of *S. apiana*, imported from the USA, were purchased from Deesis (Warsaw, Poland). The chemical composition of the obtained essential oils, determined by GC/FID and GC/MS, was originally reported in a previous study [4]. All essential oils were refrigerated in the dark at 8 °C.

### 4.2. Elicitors Preparation

Reagents for the in vitro plant culture experiments were obtained from Sigma-Aldrich (St. Louis, MO, USA), while ultrapure water was produced using the Elix/Synergy system (Merck KGaA, Darmstadt, Germany).

A 10 mg/mL stock solution of chitosan was prepared by dissolving the substance in 5% (*v*/*v*) 1 M HCl (POCH, Zabrze, Poland) by heating and stirring, followed by pH adjustment to 5.0 with 1 M NaOH. The solution was autoclaved (121 °C, 0.1 MPa, 22 min). The final chitosan concentration in the growth medium was 200 mg/L. The control sample consisted of the solvent (1 M NaCl), neutralized to pH 5.0 using 1 M NaOH.

A 2.0 mM stock solution of ergosterol was prepared by dissolving the reagent in boiling 95% (*v*/*v*) ethanol (Polmos, Siedlce, Poland). The solution was aseptically filtered (0.22 µm pore size, Chemland, Stargard, Poland). The final ergosterol concentration was 10 µM. The control sample consisted of sterile 95% (*v*/*v*) ethanol.

Yeast extract was dissolved in ultrapure water to prepare aqueous elicitor solutions (10.0 and 20.0 mg/mL), which were then autoclaved (121 °C, 0.1 MPa, 22 min). The prepared solutions were added to the RITA^®^ bioreactor at 1.0 mL, achieving final concentrations of 50.0 and 100.0 mg/L in the medium. The control was *S. apiana* culture, grown in a RITA^®^ bioreactor without any additives.

### 4.3. Elicitation Experiments in Bioreactors

The temporary immersion reactor RITA^®^ (200 mL of working volume) was used to perform elicitation experiments. The RITA^®^ reactor was steam-sterilized at 121 °C and 0.1 MPa for 22 min, then connected to an air humidifier and an air pump (IPX4 ACO-9602, Hailea, Raoping Xian, China) through a filter disk made of polytetrafluoroethylene (PTFE) with a 0.22 μm pore size and a 60 mm diameter (Cole Parmer, Vernon Hills, IL, USA). The pressure of sterile air, applied to the bottom part, was at a 500 mL/min rate. The microshoots resting on a strainer were immersed for 5 min. every 1.5 h. All plant cultures were kept at 24 ± 2 °C, under white fluorescent light (16/24 h photoperiod, 88.8 μmol/m^2^·s, TLD 35 W/33 tubes, Philips, Amsterdam, The Netherlands).

The bioreactors were inoculated at a 1:23.5 biomass to medium (*m*/*v*) ratio (ca. 8.5 g per RITA^®^ vessel). The microshoots were maintained in liquid SH medium, supplemented with 0.22 mg/L TDZ, 2.0 mg/L 2iP, and 3.0% (*w*/*v*) sucrose. All elicitors were added on the 14th or 18th day of the experiment, with exposure periods of 3 or 7 days, respectively, lasting until the 21st day of cultivation. The elicitors were introduced into the system either by membrane filtration using syringe filters (PTFE, 0.2 μm pore size, 25 mm diameter, Cronus, Maisemore, UK) or, for pre-autoclaved elicitors, by direct pipetting through the central tube of the RITA^®^ bioreactor into the liquid medium at the bottom, after the detachment of the silicone aeration hose. The procedure for adding elicitors has been thoroughly described in a paper concerning the study of *Ledum palustre* [8]. On the last day of the experiment, the microshoots were collected and assessed for growth parameters (FW, Gi, DW) and morphological features. The dried biomasses were evaluated for essential oil content and the isolated volatile fractions were subjected to GC analysis. Each experiment was conducted with at least three replicates.

### 4.4. Determination of Growth Parameters

The microshoots of *S. apiana* obtained from the bioreactor experiments were removed from the liquid medium and rinsed with distilled water. The fresh weight (*FW*) of the plant material was measured, and the growth index (*Gi*) was calculated using the following formula:*Gi *= (*FW_X_* − *FW*_0_)/*FW*_0_ × 100
where *Gi* represents the growth index, *FW*_0_ is the initial fresh weight of the inoculum, and *FW_X_* is the fresh weight of the microshoots after *X* days of cultivation. To measure dry weight (*DW*), the plant samples were dried for 24 h at 30 °C in a forced convection oven (FD 115, Binder, Tuttlingen, Germany).

### 4.5. Determination of Essential Oil Content

Dried plant samples obtained from the bioreactor experiments were subjected to hydrodistillation using a Clevenger apparatus (400 mL distilled water, 3 h) in accordance with the European Pharmacopoeia guidelines. Each distillation was performed with a minimum of 6.0 g of dry material. Volatile fractions obtained from the bioreactor experiments were diluted with 0.5 mL of xylene (Sigma-Aldrich, St. Louis, MO, USA) for analysis, whereas essential oils intended for biological tests were isolated as pure samples without xylene. All essential oils were dried over anhydrous sodium sulfate for 24 h and subsequently stored at 8 °C until GC analysis. The reported volatile oil contents represent the average values from at least three independent hydrodistillations.

### 4.6. Analysis of Essential Oils Using GC/FID and GC/MS

Qualitative analysis of *S. apiana* essential oil samples, obtained from the bioreactor experiments, was performed using gas chromatography coupled with a mass selective detector (GC/MS) on a 7890A gas chromatograph and a 5977A mass selective detector (EIMS). Quantitative analysis was carried out using a GC/FID system equipped with a flame ionization detector (5977A gas chromatograph; Agilent Technologies, Santa Clara, CA, USA). Analysis was performed using the Agilent MassHunter software (https://www.agilent.com.cn/en/promotions/masshunter-mass-spec (accessed on 5 February 2025), Santa Clara, CA, USA). Prior to analysis, the EO samples (10.0 μL) were diluted with acetone at a 1:80 (*v*/*v*) ratio.

For GC/MS analysis, the diluted samples were injected into a DB-5 ms capillary column (30 m × 0.25 mm × 0.25 μm, Agilent J&W) using a split/splitless injector (model 7693, Agilent) at a split ratio of 1:10. The injection volume was 1 μL, and the injection temperature was set at 250 °C. Helium was used as the carrier gas at a flow rate of 1.1 mL min^−1^. The oven temperature was programmed to increase from 50 °C to 280 °C at a rate of 7 °C min^−1^, with a final hold at 280 °C for 20 min.

For GC/FID analysis, a DB-5 capillary column (30 m × 0.32 mm × 0.25 μm) was used with the same oven temperature program and injector parameters as in the GC/MS analysis. The helium carrier gas flow rate for GC/FID was 1.5 mL/min. Retention indices and mass spectra from the NIST Library 11.0 were used to compare and identify the obtained data.

### 4.7. Lymphocyte Proliferation and Survival/Death Study

A group of five prescreened healthy volunteers (3 men and 2 women, mean age 35 years) were recruited. None of them were taking any medication affecting the immune system. All participants were informed of the purpose of the experiment and gave written consent. The Local Independent Research Ethics Committee at the Medical University of Gdansk approved the project (permission no. NKBBN/999/2021-2022).

Peripheral blood mononuclear cells (PBMCs) were obtained from fasting venous blood samples. Blood was collected into Vacutainer^TM^ tubes containing EDTA (BD Pharmingen, San Diego, CA, USA), and PBMCs were then isolated by Histopaque^TM^ flotation (Sigma-Aldrich, St. Louis, MO, USA). Isolated PBMCs were supravitally stained with Violet Proliferation Dye 450 (VPD450) (Becton Dickinson, East Rutherford, NJ, USA) for 10–15 min in the dark at 37 °C according to Witkowski’s protocol [47], resuspended in complete culture medium (RMPI 1640) supplemented with 10% fetal bovine serum, 2 mM glutamine, 100 U/mL penicillin, and 100 μg/mL streptomycin (all reagents from Sigma-Aldrich, St. Louis, MO, USA) at 2 × 106 mL^−1^, and stimulated with immobilized anti-CD3 (BD Pharmingen, San Diego, CA, USA).

The cells were then treated with essential oils isolated from *Salvia apiana*, and 1,8-cineol (Sigma-Aldrich, St. Louis, MO, USA), at three dilutions (1:400, 1:4000, and 1:40,000 *v*/*v*). Oils were used as stock solutions in ethanol, so the final solvent concentration in the culture medium was 0.25%. Oil-treated samples and ethanol-only controls were incubated for 72 and 120 h at 37 °C in 5% CO_2_/95% air in a humidified incubator. After incubation, cells were collected, washed with phosphate-buffered saline (PBS), and stained with the following antibodies: anti-CD3conjugated with fluorescein-5-isothiocyanate (FITC/anti-CD3), anti-CD8 conjugated with phycoerythrin (PE/anti-CD8), and anti-CD4 conjugated with a red-emitting tandem fluorophore that combines phycoerythrin and Cy5 (RPE-Cy5/anti-CD4) (all from BD Pharmingen, USA). They were then analyzed by flow cytometry according to the staining procedure described above.

### 4.8. Flow Cytometry Analysis of Lymphocyte Proliferation and Survival/Death

Thirty thousand events corresponding to lymphocytes’ light scatter characteristics were acquired from each sample to analyze proliferation capacity and cell susceptibility to apoptosis and necrosis. We used the FACSVerse™ cytometer (Becton Dickinson) to acquire data and FCSalyzer (copyright (C) 2012–2019 Sven Mostbock) to perform cytometric analysis. First, the lymphocytes were selected based on forward and side scatter characteristics (FSCs and SSCs) and their positivity for surface antigens (CD3, CD4, CD8). Then, the dividing cell tracking (DCT) method was used to determine the percentages of dividing cells after different simulation combinations. DCT uses VPD450, which passively diffuses across cell membranes and is cleaved by esterase activity within viable cells. The cleaved dye becomes fluorescent and covalently binds to proteins within the cells. As viable cells divide, the VPD450 dye is distributed uniformly between daughter cells, so each daughter cell retains approximately half of the VPD450 fluorescence intensity of its parent cell; non-dividing cells remain maximally bright which allows for their easy distinction in the flow cytometric graphs. Finally, based on annexin V and 7-AAD staining, we identified cells as alive, such as cells negative for annexin V and 7-AAD, in early apoptosis, annexin V-positive 7-AAD-negative cells, in late apoptosis, such as cells positive for both annexin V and 7-AAD, and necrotic cells—cells that were only 7-AAD-positive.

### 4.9. Statistical Analysis

Statistical data analysis was performed using the GraphPad Prism program, version 9 (GraphPad Software, San Diego, CA, USA). Shapiro–Wilk and Kolmogorov–Smirnov tests were used to test for normal distribution. Since data did not pass the normality tests, an appropriate nonparametric test for repeated measures (Friedman’s ANOVA with Dunn’s post hoc test) was chosen with a significance level of *p* < 0.05. Values are shown either in the box-and-whisker diagram with marked means, standard errors (SEs), and standard deviations (SDs) or as bars (means) and standard errors (SEs, whiskers).

## 5. Conclusions

Essential oils isolated from *S. apiana* leaves of field-grown plant and microshoot cultures of white sage, as well as 1,8-cineole, decreased the proportions of proliferating CD4+ and CD8+ T lymphocytes, simultaneously increasing the proportion of these cells undergoing late apoptosis and necrosis. Essential oils obtained from microshoots seem to be marginally more effective than oils obtained from field-grown plants. The microshoot culture of *S. apiana* appeared to exhibit resistance to elicitation techniques, while showing a modest enhancement in the capacity to accumulate volatile fractions in response to the addition of yeast extract as an elicitor.

## Figures and Tables

**Figure 1 molecules-30-00815-f001:**
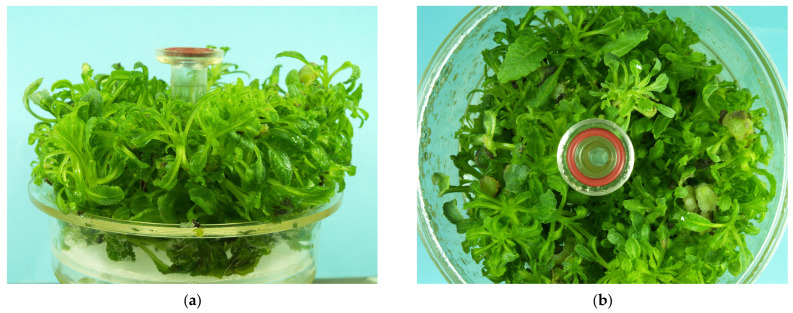
Microshoots of *S. apiana,* grown in the RITA^®^ bioreactor for 21 days, subjected to elicitation with yeast extract (100 mg/L) 7 days before harvesting: (**a**) shoot tray removed, side view; (**b**) upper view, with lid removed.

**Figure 2 molecules-30-00815-f002:**
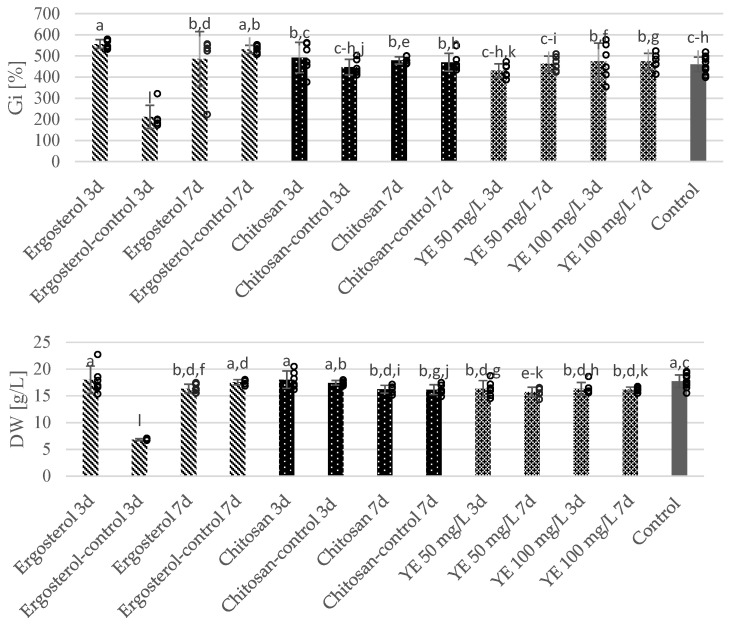
The effect of elicitation with ergosterol, chitosan and yeast extract (YE) on biomass growth in bioreactor-grown microshoots of *S. apiana*. The liquid growth medium was modified by adding elicitors or their respective solvents (positive control) either 3 days (3 d) or 7 days (7 d) before harvesting. The negative control consisted of biomass cultivated in non-elicited medium for 21 days. The presented values represent arithmetic means from a minimum of six experimental replicates ± standard deviation (SD). Dots represent individual replicates. Different letters indicate significant differences at *p* = 0.05 (one-way ANNOVA followed by Fisher LSD test).

**Figure 3 molecules-30-00815-f003:**
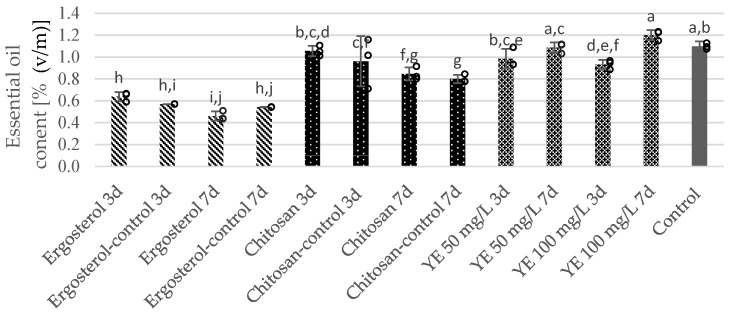
The effect of elicitation with ergosterol, chitosan and yeast extract (YE) on essential oil content in bioreactor-grown microshoots of *S. apiana*. The liquid growth medium was modified by adding elicitors or their solvents (positive controls) either 3 days (3 d) or 7 days (7 d) before harvesting. The negative control consisted of biomass cultivated in non-elicited medium for 21 days. The presented values represent arithmetic means of three experimental replicates ± standard deviation (SD). Dots represent individual replicates. Different letters indicate significant differences at *p* = 0.05 (two-way ANNOVA followed by Fisher’s LSD test).

**Figure 4 molecules-30-00815-f004:**
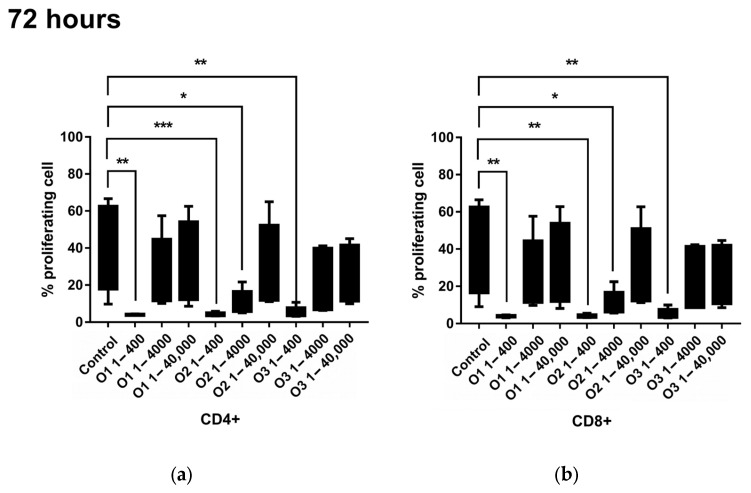
Comparison of percentage of proliferating CD4+ (**a**,**c**) and CD8+ (**b**,**d**) cells, stimulated for 72 h (**a**,**b**) and 120 h (**c**,**d**) with different dilutions of essential oils (1:400, 1:4000, and 1:40,000 *v*/*v*) isolated from *S. apiana* leaves of field-grown plants (O1) and microshoots (O2), as well as 1,8-cineole (O3). The graphs show the percentiles with the maximum and minimum value (Friedman’s ANOVA with Dunn’s post hoc test; * *p* < 0.05, ** *p* < 0.01, *** *p* < 0.001).

**Figure 5 molecules-30-00815-f005:**
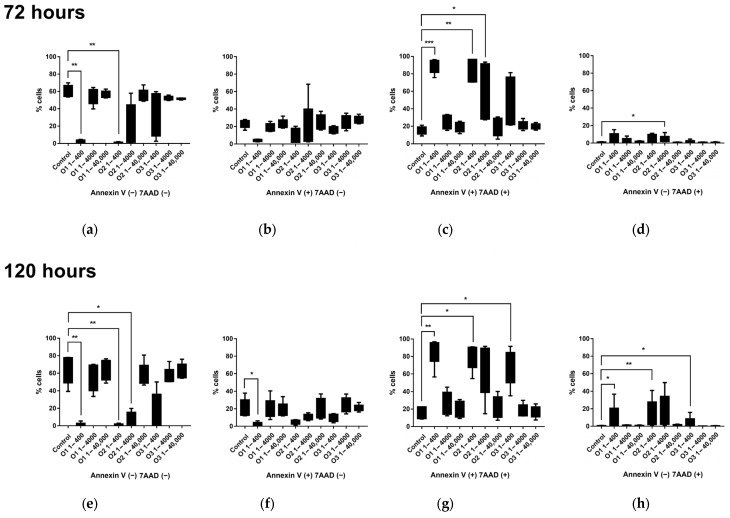
Comparison of percentage of living lymphocytes (**a**,**e**), those in early apoptosis (**b**,**f**), those in late apoptosis (**c**,**g**) and necrotic (**d**,**h**) lymphocytes stimulated for 72 h (**a**–**d**) and 120 h (**e**–**h**) with different dilutions of essential oils (1:400, 1:4000, and 1:40,000 *v*/*v*) isolated from *S. apiana* leaves of field-grown plants (O1) and microshoots (O2), as well as 1,8-cineole (O3). The graphs show percentiles with maximum and minimum values (Friedman’s ANOVA with Dunn’s post hoc test, * *p* < 0.05, ** *p* < 0.01, *** *p* < 0.001).

**Table 1 molecules-30-00815-t001:** Composition of essential oil samples obtained from leaves of field-grown plants and microshoot cultures of *S. apiana*. Values represent the % of EO, according to GC-FID.

Compound *	Rt [min]	Leaves of Field-Grown Plant	Agar-Grown Culture	RITA^®^-Grown Microshoots												
				Control (Non-Elicited Culture)	Elicitation with Ergosterol (10 µM)				Elicitation with Chitosan (200 mg/L)				Elicitation with Yeast Extract (YE)			
					3 Days Exposure	Control (EtOH) 3 Days Exposure	7 Days Exposure	Control (EtOH) 7 Days Exposure	3 Days Exposure	Control (HCl) 3 Days Exposure	7 Days Exposure	Control (HCl) 7 Days Exposure	YE 50 mg/L 3 Days Exposure	YE 50 mg/L 7 Days Exposure	YE 100 mg/L 3 Days Exposure	YE 100 mg/L 7 Days Exposure
3-thujene	5.589	ND	ND	0.403	0.397	0.345	0.402	0.438	0.379	0.396	0.424	0.402	0.425	0.391	0.418	0.372
*α*-pinene	5.701	6.701	9.649	9.152	10.006	9.781	11.374	10.281	9.102	9.286	9.164	8.519	9.798	9.150	9.400	10.454
camphene	5.945	0.394	0.402	0.332	0.366	0.364	0.504	0.380	0.334	0.341	0.314	0.307	0.322	0.303	0.350	0.333
*β*-phellandrene	6.107	ND	ND	0.408	0.513	0.533	0.513	0.491	0.503	0.487	0.448	0.440	0.566	0.597	ND	0.544
*β*-pinene	6.408	5.550	13.062	12.413	14.673	15.150	15.837	15.152	13.018	13.473	12.590	12.171	14.216	14.756	14.361	14.641
*β*-myrcene	6.523	2.191	1.890	1.683	1.352	1.144	1.344	1.508	1.464	1.474	1.827	1.628	1.667	1.678	1.716	1.522
*α*-phellandrene	6.864	ND	ND	0.150	0.115	0.087	0.110	0.125	0.124	0.126	0.159	0.144	0.159	0.146	0.165	0.140
*δ-3*-carene	6.936	2.653	1.034	1.178	1.042	0.933	1.112	1.185	1.159	1.330	1.282	0.224	1.500	1.353	1.374	1.315
*α*-terpinene	7.037	ND	0.451	0.239	0.224	0.202	0.201	0.251	0.234	0.247	0.260	0.248	0.269	0.271	0.270	0.244
limonene	7.255	0.715	0.613	0.217	0.196	0.163	0.175	0.220	0.178	0.150	0.248	0.213	0.075	0.096	0.128	0.117
1,8-cineole	7.384	72.744	50.125	52.065	35.471	28.759	37.410	38.265	40.711	42.023	48.396	47.548	44.124	44.320	44.354	38.521
*γ*-terpinene	7.785	0.561	0.848	0.683	0.597	0.572	0.589	0.630	0.627	0.639	0.725	0.669	0.687	0.686	0.709	0.636
*β*-*cis*-terpineol	8.344	ND	ND	0.169	0.123	0.109	0.131	0.127	0.148	0.137	0.162	0.169	0.148	0.164	0.155	0.127
camphor	9.481	1.112	1.634	1.723	1.427	1.452	1.697	1.572	1.567	1.551	1.483	1.591	1.433	1.396	1.427	1.396
terpinen-4-ol	10.063	0.595	0.223	0.306	0.164	0.131	0.145	0.180	0.183	0.212	0.254	0.235	0.218	0.204	0.218	0.249
*α*-terpineol	10.294	0.361	ND	0.371	0.294	0.255	0.290	0.310	0.317	0.296	0.388	0.343	0.346	0.329	0.370	0.317
*γ*-elemene	14.799	0.025	0.253	0.144	0.505	0.697	0.342	0.355	0.416	0.381	0.211	0.242	0.376	0.473	0.415	0.559
caryophyllene	16.644	0.479	3.921	3.625	9.070	11.007	7.373	7.303	7.522	6.455	3.955	4.831	4.762	4.519	4.304	6.287
aristolene	15.001	0.103	3.200	2.905	5.479	6.420	4.587	4.887	4.924	4.506	3.426	3.850	4.030	4.285	4.157	4.987
*β*-guaine	15.112	0.025	0.461	0.387	0.747	1.001	0.635	0.694	0.677	0.619	0.310	0.511	0.511	0.577	0.535	0.660
*α*-humullene	15.228	0.035	0.246	0.345	0.774	1.013	0.623	0.678	0.682	0.593	0.382	0.454	0.452	0.455	0.428	0.576
*α*-muurolen	15.598	0.040	0.245	0.534	1.034	1.299	0.825	0.964	0.919	0.830	0.629	0.712	0.705	0.724	0.691	0.829
cadinane	16.038	0.182	1.031	0.123	0.255	0.284	0.113	0.189	0.211	0.211	0.146	0.159	0.219	0.228	0.238	0.292
eudesmene	16.056	ND	ND	0.111	0.210	0.246	0.175	0.194	0.183	0.164	0.129	0.148	0.145	0.150	0.169	0.163
*β*-bisabolene	16.087	0.099	0.260	0.698	2.517	2.871	1.969	2.226	2.296	1.883	1.353	0.771	1.526	1.700	1.770	1.901
*γ*-selinene	16.593	0.034	0.243	0.136	0.217	0.290	0.183	0.220	0.226	0.187	0.151	0.169	0.153	0.152	0.143	0.171
4-epi-cubebol	16.709	0.087	0.165	0.134	0.184	0.245	0.167	0.178	0.194	0.157	0.149	0.166	0.140	0.120	0.122	0.159
cubedol	17.185	0.106	0.084	0.156	0.245	0.238	0.246	0.192	0.224	0.173	0.184	0.189	0.160	0.146	0.159	0.183
caryophyllene oxide	17.423	0.174	0.258	0.464	0.750	0.494	0.661	0.674	0.590	0.465	0.467	0.559	0.279	0.191	0.225	0.353
α-bisabolol	18.886	0.498	2.132	2.060	3.486	3.931	2.867	3.073	3.492	2.978	2.501	2.713	2.713	2.606	2.833	3.260
ferruginol	27.557	0.087	1.914	1.302	1.546	2.179	2.322	2.185	2.223	2.225	2.247	2.189	2.553	2.720	2.763	2.219
TOTAL		95.551	94.344	94.616	93.979	92.195	94.922	95.127	94.827	93.995	94.364	92.506	94.676	94.885	94.368	92.525
Monoterpene hydrocarbons		18.765	27.949	26.858	29.481	29.274	32.161	30.661	27.122	27.949	27.441	24.965	29.684	29.426	28.892	30.316
Oxygenated monoterpenes		74.812	51.982	54.634	37.479	30.706	39.673	40.454	42.926	44.219	50.683	49.878	46.268	46.413	46.524	40.609
Sesquiterpene hydrocarbons		1.022	9.860	9.008	20.808	25.128	16.825	17.710	18.056	15.829	10.692	11.847	12.879	13.263	12.850	16.427
Oxygenated sesquiterpenes		0.952	4.553	2.814	4.665	4.908	3.941	4.117	4.500	3.773	3.301	3.627	3.292	3.064	3.339	3.955

ND—not detected. * Identification based on GC-FID and GC-MS.

## Data Availability

The data presented in this study are available on request from the corresponding author due to privacy reasons (patients’ data).

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
