# Peer review of "Elicited Production of Essential Oil with Immunomodulatory Activity in Salvia apiana Microshoot Culture"

_molecules, 2025, doi:10.3390/molecules30040815_

Round 1
Reviewer 1 Report
Comments and Suggestions for Authors
This manuscript provides interesting information about the phytochemical composition of essential oils in cell cultures and shoots of a medicinal plant. The work certainly has biotechnological significance, but from a phytochemical point of view, the information provided is quite banal.
Below are some specific comments on the text of the manuscript.
- Line 35: “Salvia genus” change to “Salvia L. genus (Lamiaceae)”
- Fig. 2: what do the circles in this diagram mean?
- Fig. 3: what do the circles in this diagram mean?
- Lines 121-148, Fig 3: what do the percentage concentrations (from dry/fresh weight of biomass) mean?
- Line 161: what does "GC-MS2.2" mean?
- Lines 169, 433: “1,8-cineol” change to “1,8-cineole”.
- Table 1: Why is there no statistical information in Table 1? In general, this table is difficult to analyze (maybe the data should be presented in the form of diagrams or heat maps?).
- Throughout the text (lines 161, 163 etc.): the name of the plant species should be italicized.
- Line 326: “Salvia apiana” change to “Salvia apiana Jepson”.
- Lines 349-350: What was the final concentration of ethanol in the growing media? Perhaps the toxic concentration of ethanol in the growing medium caused a sharp drop in growth characteristics in the experimental variant "Control (EtOH) 3 days exposure" (Figure 2)? In this case, it is incorrect to compare this variant with other samples.
- Lines 348-349: It is necessary to indicate the brand and manufacturer of the filters.
Reviewer 2 Report
Comments and Suggestions for Authors
Please see the attached document for comments

Round 2
Reviewer 1 Report
Comments and Suggestions for Authors
I have no more questions. The authors made the necessary corrections to the manuscript.